

# A review and meta-analysis of the enemy release hypothesis in plant–herbivorous insect systems

Kim Meijer[1,2], Menno Schilthuizen[1,3], Leo Beukeboom[1] and Christian Smit[1]

[1] Groningen Institute for Evolutionary Life Sciences, University of Groningen, Groningen, the Netherlands
[2] Altenburg & Wymenga Ecological Consultants, Veenwouden, the Netherlands
[3] Endless Forms group, Naturalis Biodiversity Center, Leiden, the Netherlands

## ABSTRACT

A suggested mechanism for the success of introduced non-native species is the enemy release hypothesis (ERH). Many studies have tested the predictions of the ERH using the community approach (native and non-native species studied in the same habitat) or the biogeographical approach (species studied in their native and non-native range), but results are highly variable, possibly due to large variety of study systems incorporated. We therefore focused on one specific system: plants and their herbivorous insects. We performed a systematic review and compiled a large number (68) of datasets from studies comparing herbivorous insects on native and non-native plants using the community or biogeographical approach. We performed a meta-analysis to test the predictions from the ERH for insect diversity (number of species), insect load (number of individuals) and level of herbivory for both the community and biogeographical approach. For both the community and biogeographical approach insect diversity was significantly higher on native than on non-native plants. Insect load tended to be higher on native than non-native plants at the community approach only. Herbivory was not different between native and non-native plants at the community approach, while there was too little data available for testing the biogeographical approach. Our meta-analysis generally supports the predictions from the ERH for both the community and biogeographical approach, but also shows that the outcome is importantly determined by the response measured and approach applied. So far, very few studies apply both approaches simultaneously in a reciprocal manner while this is arguably the best way for testing the ERH.

Corresponding author
Christian Smit, c.smit@rug.nl

# INTRODUCTION

Understanding what determines the success of non-native species in natural environments is a key theme in biological invasion research. The Enemy Release Hypothesis (ERH) (*Williamson, 1996*) forms an important explanation for the success of non-native species: they suffer less from natural enemies (predators, parasites and herbivores) than native species for two main reasons: (1) non-native species may have been introduced without their

natural enemies, and (2) potential enemies may not yet have colonized and/or adapted to the novel species. These predictions from the ERH have been tested many times in both animals and plants, using either the community approach (studying native and non-native species in the same habitat or community) or the biogeographical approach (studying the same species in their native and introduced habitat). However, the outcomes of these studies are highly variable (*Keane & Crawley, 2002*; *Jeschke et al., 2012*) and sometimes even contradicting each other. For instance, *Colautti et al. (2004)* showed that most of the reviewed studies (15 out of 25) found support for the ERH, while six found no support and five found results opposite to the prediction of the ERH (one study found both support for and against the ERH). They therefore conclude that there is no simple relationship between enemy release and the success of non-native species.

Possibly, the contradictory results are due to the large variety of study systems incorporated. If so, one would expect less variability when focussing on a more specific system. Indeed, two reviews with such a narrower focus provided more robust results with general support for the ERH. *Torchin et al. (2003)* reviewed the parasite load of 26 animal host species (including molluscs, crustaceans, fishes, birds, mammals, amphibians and reptiles), using the biogeographical approach, and found twice as many parasite species on animals in their native habitat, compared to the introduced habitat. *Liu & Stiling (2006)* reviewed ERH studies in plant–herbivore insect systems and similarly found clear support for the ERH. However, these latter results should be interpreted with care for two reasons. First, the studies reviewed only investigated a limited number of plant species. Second, data for this review were obtained using very different methods (ranging from limited anecdotal observations to many years of field data). Hence, for a proper review of the ERH one would best focus on studies of a specific system, which apply comparable study approaches and methodologies, and with sufficiently large numbers of species studied.

We performed a review of the ERH using one type of study system, namely plants and their insect herbivores. We compiled a large number (68) of datasets that use the community or biogeographic approach, incorporating in total more than 700 plant species. With this dataset we performed a meta-analysis for both approaches to test the predictions of the ERH that the insect diversity, insect load and level of herbivory are higher on native host plants (or in native habitats) than on non-native host plants (or in non-native habitats).

## MATERIALS AND METHODS

### Data collection

The two main methods used to study the effect of the ERH—the community approach and the biogeographical approach—each have their benefits and disadvantages. The benefit of the community approach (studying different native and non-native species in the same community) is that all species are studied within the same environment. However, the disadvantage is that taxonomically very different plant species are compared, often even from different families. Since the diversity and numbers of herbivorous insects (hereafter insects) between plant taxa may vary greatly (*Agrawal & Kotanen, 2003*), the choice of plants studied can greatly influence the outcome of study. Probably the best way to use the community

approach is to compare native and introduced species from the same genus, as some studies have done (*Auerbach & Simberloff, 1988*; *Kennedy & Southwood, 1984*; *Leather, 1986*; *Proches et al., 2008*; *Sugiura, 2010*), or to encompass a substantial representation of a region's entire flora. With the biogeographical approach (comparing the same plant species in both native and novel habitat) this problem of using taxonomic different plant species is avoided, but the community in which the species are studied may be very different. From here onwards, when considering the biogeographical approach, we will refer to *native species* for species growing in their native area and to *non-native species* for species growing in their introduced area. In line with the reviewed papers, we here use a rather broad definition of success of non-native species that includes species that successfully establish and (rapidly) expand in the novel environment, often—but not necessarily—becoming pest species (i.e., resulting in large negative economic or ecological impact (sensu *Williamson, 1996*)).

We searched Web of Science for relevant studies using single or a combination of the following keywords: native, non-native, insect, plant, introduced, non-indigenous, exotic, invasive. Furthermore, we used cross-referencing to find additional articles referring to these retrieved studies. In total, we found 68 datasets in 44 articles published between 1974 and 2015, studying one of the following variables: number of insect species (measure of diversity, 33 datasets, Table 1), number of insect individuals (measure of insect load, 18 datasets, Table 2) and level of herbivory—predominantly scored as number/percentage of leaves with herbivore damage (including leaf mining), percentage herbivore damage per leaf, or differences in leaf biomass (17 datasets, Table 3). For all datasets, the means and standard deviations of the above-mentioned variables were taken (if only standard error was given, standard deviation was calculated based on standard error and sample size), as well as the number of plant species for native and non-native plant species. We excluded eight datasets from the analyses on the number of insect species (*Clement, 1990*; *Jobin, Schaffner & Nentwig, 1996*; *Johnson, Mccaffrey & Merickel, 1992*; *Lindelöw & Björkman, 2001*; *Moran, 1980*; *Morrow & Lamarche, 1978*; *Strong, Lawton & Southwood, 1984*; *Tewksbury et al., 2002*; *Waloff, 1966*), as in these eight studies the data collection of the insects on native plants was done during a different study and/or using a different method than the data collection of the insects on non-native plants. One of the datasets (Table 2, dataset 42) was collected by one of the authors (KM) in 2009 and is here reported for the first time (K Meijer, 2009, unpublished data). Native and non-native plant species were sampled for herbivore insects in three nature areas in the North of the Netherlands that are quite similar in tree species composition and land-use history, and within a 15 km radius (Noordlaarder Bos 53.117°N, 6.644°E; Vosbergen 53.141°N, 6.583°E; Mensingebos 53.123°N, 6.436°E). Eight native plants (*Betula pendula*, *Frangulus alnus*, *Lonicera periclymenum*, *Populus alba*, *Prunus avium*, *Prunus padus*, *Quercus robur*, *Sorbus aucuparia*) and four non-native plant species (*Amelanchier* spec., *Prunus serotina*, *Quercus rubra*, *Robinia pseudoacacia*) were sampled. To standardize data collection, we selected low hanging branches (up till 2 m high) of at least 1 m long that were shaken for 10 s above a 1 m$^2$ sheet, and counted the number of herbivore insects in the sheet. All other collected functional groups such as predatory or omnivorous insects were discarded. Samples were taken on average 8.25 times per plant species.
**Table 1  Overview of datasets on the differences in insect diversity between native and non-native plants using the community (1–22) or the biogeographical approach (23–33).** Indicated are for each dataset the plant sample size (number of native/non-native plant species), the mean difference in insect diversity (number of species), and the standardized mean difference in insect diversity (±95% confidence interval). Overall outcome of random effects (RE) model indicated in italics. Significant differences are indicated in bold.

| | Dataset and reference | Sample size (native/non-native plants) | Mean difference insect diversity (native–non-native) | Standardized mean diff. insect diversity (±95% CI) |
|---|---|---|---|---|
| **Community approach** | | | | |
| 1 | *Agrawal et al. (2005)* | 14/14 | 0.11 | 0.08 [−0.66, 0.82] |
| 2 | *Auerbach & Simberloff (1988)* | 1/2 | 4.00 | |
| 3 | *Bürki & Nentwig (1997)* | 1/1 | 0.00 | |
| 4 | *Engelkes et al. (2012)* | 2/2 | 36.60 | 0.68 [−1.33, 2.70] |
| 5 | *Goßner, Gruppe & Simon (2005)* | 1/1 | −5.00 | |
| 6 | *Goßner, Liston & Späth (2007)* | 4/2 | 0.07 | 0.48 [−1.24, 2.20] |
| 7 | *Hartley, Rogers & Siemann (2010)* | 3/1 | 6.61 | |
| 8 | *Harvey et al. (2015)* | 5/5 | 0.01 | 0.05 [−1.19 , 1.29] |
| 9 | *Helden, Stamp & Leather (2012)* | 16/7 | 1.39 | 0.51 [−0.39, 1.41] |
| 10 | *Jobin, Schaffner & Nentwig (1996)* | 1/1 | 33.00 | |
| 11 | *Kennedy & Southwood (1984)* | 21/7 | 12.62 | **1.08 [0.18, 1.99]** |
| 12 | *Leather (1986)* | 46/13 | 14.11 | 0.44 [−0.19, 1.06] |
| 13 | *Liu & Stiling (2006)* | 2/4 | 4.00 | 1.38 [−0.49, 3.24] |
| 14 | *Meijer et al. (2015)* | 8/20 | 0.31 | 0.79 [−0.06 , 1.63] |
| 15 | *Meijer et al. (2015)* | 19/19 | 0.40 | 0.50 [−0.15 , 1.14] |
| 16 | *Novotny et al. (2003)* | 1/2 | −0.82 | |
| 17 | *Proches et al. (2008)* | 3/9 | 2.26 | 1.02 [−0.35, 2.38] |
| 18 | *Radho-Toly, Majer & Yates (2001)* | 2/2 | 0.80 | 1.77 [−0.54, 4.08] |
| 19 | *Southwood, Moran & Kennedy (1982)* | 4/2 | 44.75 | **2.30 [0.16, 4.44]** |
| 20 | *Southwood, Moran & Kennedy (1982)* | 3/3 | 14.00 | 1.33 [−0.44, 3.10] |
| 21 | *Southwood et al. (2004)* | 2/2 | 130.5 | 1.46 [−0.75, 3.67] |
| 22 | *Sugiura (2010)* | 102/49 | 0.45 | **0.77 [0.42, 1.13]** |
| | *RE model* | *261/168* | | *0.67 [0.46 , 0.89]* |
| **Biogeographical approach** | | | | |
| 23 | *Cripps et al. (2006)* | 1 | 33.0 | |
| 24 | *Goeden (1974)* | 1 | 18.0 | |
| 25 | *Meijer et al. (2015)* | 2 | 0.96 | 0.66 [−1.35 , 2.68] |
| 26 | *Meijer et al. (2015)* | 12 | 0.31 | 0.31 [−0.49 , 1.12] |
| 27 | *Memmott et al. (2000)* | 2 | 11.0 | **3.10 [0.19, 6.01]** |
| 28 | *Moore (1975)* | 1 | 2.0 | |
| 29 | *Southwood, Moran & Kennedy (1982)* | 2 | 51.5 | **4.64 [0.87, 8.40]** |
| 30 | *Southwood et al. (2004)* | 1 | 8.0 | |
| 31 | *Szentesi (1999)* | 1 | 9.0 | |
| 32 | *Wilson & Flanagan (1990)* | 1 | 42.0 | |
| 33 | *Wilson, Flanagan & Gillett (1990)* | 1 | 19.0 | |
| | *RE model* | *25* | | *1.59 [−0.19–3.37]* |

**Table 2  Overview of datasets on the differences in insect load (number of insect individuals) between native and non-native plants using the community (34–47) or the biogeographical approach (48–51).** Indicated are for each dataset the plant sample size (number of native/non-native plant species), the mean difference in insect diversity (number of species), and the standardized mean difference in insect diversity (±95% confidence interval). Overall outcome of random effects (RE) model indicated in italics. Significant differences are indicated in bold.

| | Dataset and reference | Sample size (native/non-native plants) | Mean difference in insect load (native–non-native) | Standardized mean diff. (± 95% CI) |
|---|---|---|---|---|
| **Community approach** | | | | |
| 34 | *Auerbach & Simberloff (1988)* | 1/2 | 10.77 | |
| 35 | *Cincotta, Adams & Holzapfel (2008)* | 1/1 | 10.0 | |
| 36 | *Goßner, Gruppe & Simon (2005)* | 1/1 | −557.0 | |
| 37 | *Goßner, Liston & Späth (2007)* | 4/2 | 0.25 | 0.60 [−1.14, 2.33] |
| 38 | *Harvey et al. (2015)* | 5/5 | −0.04 | −0.20 [−1.45, 1.04] |
| 39 | *Helden, Stamp & Leather (2012)* | 16/7 | 2.28 | 0.60 [−0.30,1.51] |
| 40 | *Meijer et al. (2015)* | 8/20 | 1.58 | **1.10 [0.23 , 1.97]** |
| 41 | *Meijer et al. (2015)* | 19/19 | 3.79 | 0.30 [−0.34 , 0.94] |
| 42 | K Meijer, 2009, unpublished data | 8/4 | 0.57 | 0.13 [−1.07, 1.34] |
| 43 | *Novotny et al. (2003)* | 1/2 | −0.15 | |
| 44 | *Southwood, Moran & Kennedy (1982)* | 4/2 | 4,902.3 | 0.56 [−1.17, 2.28] |
| 45 | *Southwood, Moran & Kennedy (1982)* | 3/3 | 294.7 | 1.01 [−0.69, 2.71] |
| 46 | *Yela & Lawton (1997)* | 8/3 | 17.54 | 0.08 [−1.25, 1.41] |
| 47 | *Zuefle, Brown & Tallamy (2008)* and *Zuefle (2006)* | 15/30 | −106.2 | −0.46 [−1.08, 0.17] |
| | *RE model* | *91/97* | | *0.29 [−0.11 , 0.17]* |
| **Biogeographical approach** | | | | |
| 48 | *Fenner & Lee (2001)* | 13 | 4.43 | 0.67 [−0.12, 1.46] |
| 49 | *Meijer et al. (2015)* | 2 | −3.35 | −0.34 [−2.32 , 1.63] |
| 50 | *Meijer et al. (2015)* | 12 | 8.58 | 0.57 [−0.24 , 1.39] |
| 51 | *Southwood, Moran & Kennedy (1982)* | 2 | 11,090.5 | 0.96 [−1.11, 3.02] |
| | *RE model* | *29* | | *0.58 [0.05 , 1.10]* |

The data of datasets 1 and 47 could not be extracted directly from the articles (*Agrawal et al., 2005*; *Zuefle, Brown & Tallamy, 2008*). Anurag Agrawal kindly provided us with the raw data of dataset 1, and Marion Zuefle provided us with a copy of her thesis (*Zuefle, 2006*) which contained the raw data of dataset 47. We derived the data of datasets 7, 8, 38, 55, 58, 61, 65–68 from the figures in the articles (*Adams et al., 2008*; *Cripps et al., 2010*; *Vilà, Maron & Marco, 2005*; *Hartley, Rogers & Siemann, 2010*; *Harvey et al., 2015*; *Lieurance & Cipollini, 2013*; *Lombardero, Vázquez-Mejuto & Ayres, 2008*; *Wolfe, 2002*).

## Data analysis

For all datasets the mean, standard deviation, and number of plant species were taken for native and for non-native plants. The mean difference (mean of native plants/mean of non-native plants) was calculated and is shown in Tables 1–3. Then, for each group of datasets a meta-analysis was done using the R package metafor (*Viechtbauer, 2010*) (Random-effects model; model specification: Hedges estimator (Hegdes g)). Datasets which contained either only one native and/or only one non-native plant species could not be used and were excluded from the analysis. Since all datasets on the level of herbivory using the

**Table 3  Overview of datasets on the differences in level of herbivory between native and non-native insects, using the community (52–64) or the biogeographical approach (65–68).** Indicated are for each dataset the plant sample size (number of native/non-native plant species), the mean difference in insect diversity (number of species), and the standardized mean difference in insect diversity (± 95% confidence interval). Overall outcome of random effects (RE) model indicated in italics. Significant differences are indicated in bold.

| | Dataset and reference | Sample size (native/non-native plants) | Mean difference in herbivore level (native–non-native) | Standardized mean diff. (± 95% CI) |
|---|---|---|---|---|
| **Community approach** | | | | |
| 52 | *Agrawal & Kotanen (2003)* | 15/15 | −1.66 | −0.17 [−0.89, 0.54] |
| 53 | *Carpenter & Cappuccino (2005)* | 30/39 | 1.20 | **0.61 [0.12, 1.10]** |
| 54 | *Cincotta, Adams & Holzapfel (2008)* | 1/1 | −1.16 | |
| 55 | *Harvey et al. (2015)* | 5/5 | 0.08 | 1.11 [−0.22 , 2.45] |
| 56 | *Heard & Sax (2013)* | 6/6 | −17.23 | **−2.31 [−3.77, −0.85** |
| 57 | *Hill & Kotanen (2010)* | 20/15 | 0.62 | **0.77 [0.09, 1.47]** |
| 58 | *Lieurance & Cipollini (2013)* | 1/1 | 19.73 | |
| 59 | *Liu, Stiling & Pemberton (2007)* | 2/1 | 0.06 | |
| 60 | *Liu, Stiling & Pemberton (2007)* | 2/4 | 0.14 | **10.28 [4.22, 16.34]** |
| 61 | *Lombardero, Vázquez-Mejuto & Ayres (2008)* | 1/1 | 12.60 | |
| 62 | *Meijer et al. (2015)* | 6/11 | 6.55 | 0.6 [−0.41, 1.62] |
| 63 | *Meijer et al. (2015)* | 12/4 | 2.47 | 0.61 [−0.54, 1.76] |
| 64 | *Schutzenhofer, Valone & Knight (2009)* | 1/1 | 9.08 | |
| | *RE model* | *102/104* | | *0.91 [−1.89, 3.70]* |
| **Biogeographical approach** | | | | |
| 65 | *Adams et al. (2008)* | 1 | 5.02 | |
| 66 | *Cripps et al. (2010)* | 1 | 25.80 | |
| 67 | *Vilà, Maron & Marco (2005)* | 1 | 19.79 | |
| 68 | *Wolfe (2002)* | 1 | 24.00 | |

biogeographical approach studied only one plant species, it was not possible to perform a meta-analysis on these datasets. For each dataset the standardized mean difference and 95% confidence interval were calculated, and for each group of datasets the overall standardized mean differences and 95% confidence interval was calculated and tested (Tables 1–3). We report the results for the number of insect species, number of insect individuals and level of herbivory for each approach separately (community and biogeographical approach).

## RESULTS

### Number of insect species
#### Community approach
In total 22 datasets (261 native and 168 non-native plant species) were found that used the community approach to study differences between native and non-native plants in the number of insect species. Most studies (19) found more insect species on native than on non-native plant species, two found the opposite, and one found no differences. Six of the datasets consisted of only one native and/or only one non-native plant species. These datasets could not be included in the meta-analysis. The analysis of the remaining 16 datasets (256 native and 162 non-native plant species) showed significantly higher numbers

of insect species on native than on non-native plant species ($z = 6.16$, $P < 0.001$, Table 1, Fig. 1A).

### Biogeographical approach
Eleven datasets (25 plant species) were found using the biogeographical approach to study differences between plants growing in their native and non-native range. All of these datasets showed higher numbers of insect species in the native than in the non-native range, but all consisted of low numbers of plant species studied. Seven datasets studied only one plant species, the other four datasets studied two species (3 datasets) and 12 (one dataset). These last four datasets (containing more than one plant species) were used for the meta-analysis, which showed a non-significant trend that plants growing in their native range contained more insect species than when growing in their non-native range ($z = 1.75$, $P = 0.08$, Table 1, Fig. 1A).

## Number of insect individuals
### Community approach
Fourteen datasets (94 native and 101 non-native plant species) used the community approach to study the differences in the number of insect individuals between native and non-native plant species. Ten datasets showed higher numbers of insects on native than on non-native plants, four the opposite. Ten of the datasets (90 native and 95 non-native species) consisted of enough plant species to be used for the meta-analysis, which showed no significant differences in the number of insect individuals between native and non-native plant species ($z = -1.44$, $P = 0.15$, Table 2, Fig. 1B).

### Biogeographical approach
Four datasets (29 plant species) were found that used the biogeographical approach to study the differences in the number of insect individuals between plants growing in their native and non-native range. Three datasets showed higher numbers of insect individuals on plants growing in their native than their introduced range, one dataset showed the opposite result. Overall, the numbers of insect individuals was significantly higher on plants growing in their native than their introduced range ($z = 2.14$, $P = 0.03$, Table 2, Fig. 1B).

## Level of herbivory
### Community approach
Thirteen datasets (102 native and 104 non-native plant species) were found that used the community approach to study the differences in the level of herbivory between native and non-native plant species. Ten datasets found higher levels of herbivory on native than on non-native plant species, three datasets found the opposite. Eight of these datasets (96 native and 99 non-native species) could be used for the meta-analysis, showing no significant difference in the level of herbivory between native and non-native plant species ($z = 0.996$, $p = 0.32$, Table 3, Fig. 1C).

### Biogeographical approach
Four datasets were found that used the biogeographical approach to study the differences in the level of herbivory between plants growing in their native and their non-native range. All

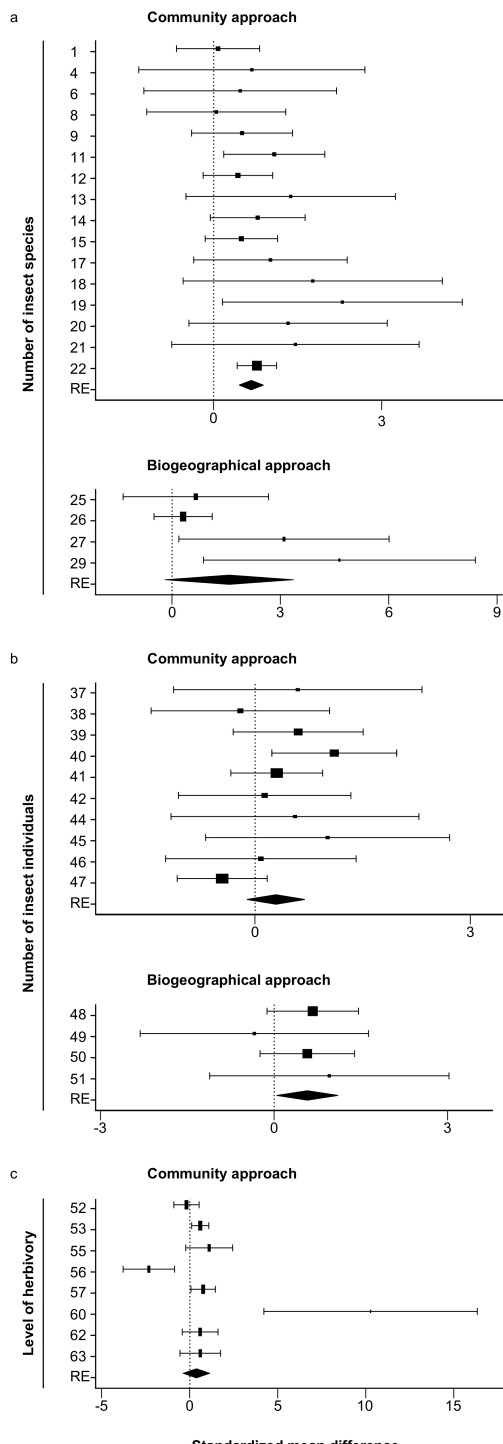

**Figure 1** **Forest plots of the standardized mean differences of the number of insect species (A), the number of insect individuals (B) and level of herbivory (C) for all studies with sample sizes >1.** The study numbers are shown at the *y*-axis, and details about the studies can be found in Tables 1–3. RE: random-effects model. Values which are lower than zero, zero or higher than zero, respectively, indicate lower, equal or higher numbers on native than on non-native plants. Diamonds not overlapping the zero line indicate significant differences of the overall random effects (RE) model.

four datasets showed higher levels of herbivory when plants were growing in their native range. Unfortunately, all four datasets consisted of only one plant species studied, and therefore it was not possible to perform a meta-analysis on these datasets.

## DISCUSSION

Regardless of the applied approach (community or biogeographical), the results of this meta-analysis indicate that native plants contain more insect species than non-native plant species do. Results for insect load and level of herbivory point in the same direction, but differences between native and non-native plants were only marginal or not significant, depending on the applied approach (community or biogeographical approach). Hence, our results are relatively consistent and, at least partly, in line with the predictions of the ERH. The narrower focus on a specific study system—here: plants and their herbivorous insects—may have helped to reduce the variability and contradictory results that others previously encountered (e.g., *Colautti et al., 2004*).

Our finding that non-native plants contained fewer herbivorous insect species than native plants can best be explained by the high level of specialization of the majority of herbivorous insects. Most herbivorous insects are extreme specialists, adapted to feeding on only one plant genus (monophage) or one plant species (strict monophage) (*Thompson, 1994*). Even the more generalist species often consist of locally specialized populations, races or even sibling species, e.g., the ermine moth (Lepidoptera) *Yponomeuta padellus* L. complex feeding on a various Rosaceae species (*Menken, 1996*). Most species of phytophagous insects appear to have speciated due to sequential evolution (*Miller & Wenzel, 1995*).

Our results show that both insect load and herbivory levels on non-native plants are not—or only marginally—different from those on native plants. In fact, it may be very advantageous for herbivorous insects, even monophages, to shift hosts from native to non-native plants as this novel environment is relatively free of competitors. Indeed, host shifts from native to non-native plants by specialist herbivorous insects have been reported. The most famous example is the host shift by the apple maggot fly *Rhagoletis pomonella* Walsh (Diptera: Tephritidae) from the native hawthorn (*Crataegus* spp.) to apple (*Malus sylvestris*) after the introduction of apple into North America some 400 years ago (*Bush, 1969*; *McPheron, Smith & Berlocher, 1988*). A more recent example is the host shift by the leaf beetle *Gonioctena quinquepunctata* from native *Sorbus aucuparia* to *Prunus serotina* in Western-Europe, leading to (weak) host-specific differentiation in the beetle populations (*Schilthuizen et al., 2016*). With the increasing number of established non-native plants it is to be expected that other native herbivorous insects will follow over time.

With time since arrival in a novel environment also the number of insect species *per* non-native plant is expected to increase. Shortly after arrival, only very few insect species will be able to shift host plant, resulting in lower numbers of insect species and/or a lower insect load (number of insects) feeding on non-native plants. However, over time, these numbers may increase, as evidenced by the study of *Brändle et al. (2008)*. Although we were not able to take into account the time since arrival of non-native plants in this study, it would be very interesting to study how this affects the outcome of such a meta-analysis.

Almost all studies we reviewed used a 'one-way' approach: studies using the community approach that compared native and non-native plants in one habitat—but not reciprocally, and studies using the biogeographical approach that compared plants in their native and non-native habitat, but—again—not in a reciprocal way. Using only one of the one-way approaches is likely to result in either over- or underestimation of the differences between native and non-native plants (*Meijer et al., 2015*). Using both approaches simultaneously in a reciprocal way is arguably the best method for testing the predictions from the ERH, but despite the numerous studies performed on the ERH, very few studies have so far applied this method.

## ACKNOWLEDGEMENTS

We would like to thank Anurag Agrawal and Marion E. Zuefle, who provided us with the raw data of their research.

### Funding

This work was supported by the Uyttenboogaart-Eliasen Foundation. The funders had no role in study design, data collection and analysis, decision to publish, or preparation of the manuscript.

### Grant Disclosures

The following grant information was disclosed by the authors:
Uyttenboogaart-Eliasen Foundation.

### Competing Interests

The authors declare there are no competing interests. Kim Meijer is an employee of Altenburg & Wymenga Ecological Consultants and Menno Schilthuizen is an employee of Naturalis Biodiversity Center.

### Author Contributions

- Kim Meijer conceived and designed the experiments, performed the experiments, analyzed the data, contributed reagents/materials/analysis tools, wrote the paper, prepared figures and/or tables, reviewed drafts of the paper.
- Menno Schilthuizen, Leo Beukeboom and Christian Smit contributed reagents/materials/analysis tools, wrote the paper, reviewed drafts of the paper.

### Data Availability

The raw data has been supplied as a Supplementary File.

### Supplemental Information

Supplemental information for this article can be found online at http://dx.doi.org/10.7717/peerj.2778#supplemental-information.

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
