# Peer review of "A review and meta-analysis of the enemy release hypothesis in plant–herbivorous insect systems"

_PeerJ, doi:10.7717/peerj.2778_

## Round 0.1 · original submission · Minor Revisions

· Academic Editor

Minor Revisions

Three top-ranked experts in the field evaluated your manuscript, suggesting minor revisions. Please address their comments in due course and submit a revised version.

·

Basic reporting

All of my comments are in the 'General Comments' section below

Experimental design

All of my comments are in the 'General Comments' section below

Validity of the findings

All of my comments are in the 'General Comments' section below

Additional comments

This paper is an update of Liu & Stiling 2006 in which the enemy release hypothesis for plants and herbivores is evaluated using the available literature. It is certainly an improvement on L & S and would add significantly to the literature on the hypothesis. I like the separation into community and biogegraphical approaches and consideration of the 3 response variables. I have some relatively minor questions on the analysis and feel that the empirical study included is not well enough explained. These and some other issues are explained below.

Abstract: first sentence – I think that saying that ERH is ‘commonly accepted’ is a straw man. That statement is followed by evidence that some studies have looked for it and not found it, which contradicts the ‘commonly accepted’ statement.

Line 61. Please specify if this is number of parasite species or individuals

Line 104. Standard deviation is not typically reported (rather standard error is). Please explain whether SD values were calculated from the data (possible if SEM and sample size are reported) or if instead SEM was used. Also – the actual values of variance (whether SD or SEM) are not always reported but rather shown in graphs as error bars. Please explain whether the values were sometimes estimated from error bars or whether published values were always used. More importantly, it was not clear to me how or if these values were used in the meta-analyses. Please be explicit about this and if they were not used, do not even mention them.

Lines 111 – 119. The description of the methods for the empirical study is too brief. Many of these plants are trees that can get very large so it is hard to imagine how the described methods could have been employed. And there is no explication of how size/age/habitat of the native and non-native plants was standardized. And I do think that the data collected (only abundance – no attempt to ID anything) is too minimalistic; there are some taxa that include both herbivores and predators and others that are omnivorous, so there are some dangers in this approach I think. I feel that this experiment has to either be much better explained or should be removed from the analysis.

Lines 132-4. I can see why community studies with only one plant species could be used, but you seem to suggest that even studies with one each native and introduced species could not be used. Am I misreading this (suggested by the ‘and/or’)? If not, I don’t see why these kinds of studies can’t be included – it would seem that the advantage of the meta-analysis approach is that these kinds of studies can be incorporated. Maybe a weighting factor could be used to take into account the number of plants used in the community analyses. Related to this I also don’t see why the meta-analysis approach is not possible the biogeographical approach, which by often uses a single species. Also, line 157 seems to discuss results from one of these disallowed studies (?). This is confusing too.

Line 141. Should the heading ‘Results’ appear here?

Line 208 and on. The authors may be interested in a paper by Andow & Imura (1994 Ecology) on this general topic.

Line 220. Do you really mean ‘shift hosts’ or are you talking about a host-range expansion? I think true shifts are much more rare than expansion of host range. The examples given that I’m aware of aren’t really true ‘shifts’ they are host-range expansions followed by host-race formation.

Reviewer 2 ·

Basic reporting

The manuscript “A review and meta-analysis of the enemy release hypothesis in plant – herbivorous insect systems” by Meijer and colleagues is relevant for people interested in invasion ecology. In general, the manuscript is clear and well written. The manuscript tests the predictions of the enemy release hypothesis (ERH). For this, authors have used the plant-phytophagous insect system and two approaches: community and biographical. Since the ERH remains equivocal, these studies are still necessary. In this sense, the introduction describes properly the current status of the ERH and the aims of this manuscript, although I miss some references (i.e. Jeschke et al. (2012) in Neobiota; or Keane and Crawley (2002) In Trends in Ecology and Evolution) and authors should mention the potential use of exclusion experiments (see Allan and Crawley (2011) in Ecol. Letters).

Experimental design

The methodology is appropriate, especially taking into consideration the potential heterogeneity of the datasets. The data is also robust for most of the variables, it includes a large number of datasets (68). Maybe, if possible, authors should explain what have considered as “level of herbivory” (line 105).
.

Validity of the findings

The discussion section is well structured and contributes to understand the result section, but some of the results overestimated. For example, there are not significant differences (line 159-61), or are marginal (P = 0.08), between the number of insect species growing in their native and non-native range.

Annotated reviews are not available for download in order to protect the identity of reviewers who chose to remain anonymous.

Reviewer 3 ·

Basic reporting

The manuscript entitled “A review and meta-analysis of the enemy release hypothesis in plant-herbivore insect systems” by Meijer et al. reviews the existing literature that tested for the enemy release hypothesis (ERH) in insect-plant systems. The authors used both a community approach (native vs non-native species studied in the same habitat) and a biogeographical approach (species studied in their native and non-native range) in their analysis. They tested the ERH for insect diversity (number of species), insect load (number of insect individuals), and level of herbivory for both approaches. In general they found support for ERH for insect diversity and for the number of insect individuals for the biogeographical approach. No strong support was found for the level of herbivory. The manuscript adheres to PeerJ standards and policies.

Experimental design

This manuscript is well-written and the information clearly presented. The introduction is well referenced and relevant. The manuscript meets the journal standards.

Validity of the findings

Although there have been other reviews on the enemy release hypothesis (e.g. Colautti et al. 2004, Torchin et al. 2003, Liu and Stiling 2006), this review focuses on plants and their insect herbivores. Although this makes the paper novel, a disadvantage of focusing on a specific system is the fact that the authors used a limited set of data to test their hypotheses and, in some cases (e.g. herbivory for biogeographical approach), the dataset was not sufficient for a meta-analysis. Regardless, I believe the authors provide useful information for our current understanding of the ERH in insect-plant systems.

Additional comments

Editorial comments:
Line 72: this dataset we
Line 110: and/or using
Line 138: were calculated
Line 228: et al., 2016
Line 232: and/or
Line 242: Meijer et al., 2015
Line 253: (italics) Acer platanoides
Line 267: (italics) Heracleum sphondylium
Line 268: (italics) H. mantegazzianum
Line 273: (italics) Acer platanoides
Line 274: (italics) A. saccharum
Line 280: (italics) Cirsium arvense
Line 283: (italics) Lepidium draba
Line 287: Applied
Line 291: (italics) Cardus pycnocephalus
Line 294: (italics) Pseudotsuga menziesii
Line 294: (italics) Picea abies
Line 313: (italics) Solidago altissima
Lines 332-333: edit reference
Line 335: (italics) Rhagoletis pomonella
Lines 336-338: edit reference
Line 340: (italics) Cytisus scoparius
Line 364: weeds:
Line 364: (italics) Cirsium arvense
Line 353: (italics) Piper aduncum
Line 354: (italics) P. umbellatum
Line 374: (italics) Aporpha fructicosa
Line 377: (italics) Phragmites australis
Line 384: (italics) Hypericum perforatum
Line 385: (italics) Sarothamnus scoparius
Line 389: (italics) Sida acuta
Line 389: (italics) Sida cordifolia
Line 392: (italics) Mimosa pigra
Table 2: RE model (Biogeographical approach), I believe should be “29” instead of “2900”

---

## Round 0.2 · accepted · Accept

· Academic Editor

Accept

The manuscript has been properly revised and can be accepted now.

·

Basic reporting

I reviewed a previous version of this paper and suggested minor revisions. The authors have satisfactorily implemented these and the paper is now ready for publication in my opinion.

Experimental design

All my comments are in the 'basic reporting' section

Validity of the findings

All my comments are in the 'basic reporting' section

Additional comments

All my comments are in the 'basic reporting' section

Reviewer 2 ·

Basic reporting

No comments

Experimental design

No comments

Validity of the findings

No comments

Additional comments

No comments

Reviewer 3 ·

Basic reporting

I read the revised manuscript and find it suitable for publication. All my comments and the comments of other reviewers were incorporated into the new version.

Experimental design

Appropriate

Validity of the findings

Appropriate

Additional comments

I find the revised manuscript suitable for publication.